# Analysis of Diagnostic Images of Artworks and Feature Extraction: Design of a Methodology

**DOI:** 10.3390/jimaging7030053

**Published:** 2021-03-12

**Authors:** Annamaria Amura, Alessandro Aldini, Stefano Pagnotta, Emanuele Salerno, Anna Tonazzini, Paolo Triolo

**Affiliations:** 1Department of Pure and Applied Sciences, University of Urbino Carlo Bo, 61029 Urbino, Italy; alessandro.aldini@uniurb.it; 2Department of Earth Sciences, University of Pisa, 56126 Pisa, Italy; stefano.pagnotta@unipi.it; 3National Research Council of Italy, Institute of Information Science and Technologies, 56124 Pisa, Italy; emanuele.salerno@isti.cnr.it (E.S.); anna.tonazzini@isti.cnr.it (A.T.); 4Department of Earth Sciences of the Environment and Life, Methodologies for Conservation and Restoration of Cultural Heritage, University of Genoa, 16126 Genova, Italy; triolox@libero.it

**Keywords:** cultural heritage, diagnostic images, image analysis, feature extraction, documentation, geographic information systems (GIS)

## Abstract

Digital images represent the primary tool for diagnostics and documentation of the state of preservation of artifacts. Today the interpretive filters that allow one to characterize information and communicate it are extremely subjective. Our research goal is to study a quantitative analysis methodology to facilitate and semi-automate the recognition and polygonization of areas corresponding to the characteristics searched. To this end, several algorithms have been tested that allow for separating the characteristics and creating binary masks to be statistically analyzed and polygonized. Since our methodology aims to offer a conservator-restorer model to obtain useful graphic documentation in a short time that is usable for design and statistical purposes, this process has been implemented in a single Geographic Information Systems (GIS) application.

## 1. Introduction

The artworks undergo profound changes in time due to several factors: the natural aging of materials, pathologies of degradation, wrong restoration or remaking that introduce new materials and chemical interactions. For this reason, any technique for detecting and reporting what is not directly visible or perceptible is an essential means of diagnostic investigation. This need is currently widely met by techniques inspired by remote sensing, such as multispectral and hyperspectral imaging, as they can provide information on the composition of materials without taking samples. The technical information related to the nature and conditions of the artwork is transcribed in the graphic documentation.

The documentation refers to the systematic collection of information derived from the diagnostic investigation, restoration, monitoring, and maintenance performed on cultural heritage. Specifically, the graphic documentation, also called the thematic map, is the primary tool for communication and synthesis of the information collected on the nature and conditions of the artwork, which are transcribed into geometrically correct drawings and translated into conventional graphic symbols [1,2]. Thematic maps are generally used by different types of professionals operating at different times and in different ways, representing the formal and unequivocal means of communication, comparison, and guidance for subsequent conservation and preservation operations. The graphic documentation should always be well archived, accessible, and usable; therefore, it should be possible to obtain or arrive at the information when and where it is needed.

In the documentation process, these graphic drawings are intended for critical analysis of data. They differ from the artifact’s photographic reproduction, which detects and reproduces all its complexities in an undifferentiated way, without a critical/interpretative filter. The vector drawing allows us to realize a process of synthesis, discrepancy, and characterization of data, making the results immediately readable and statistically analyzable. Although software for the documentation of three-dimensional models is being developed very slowly and pioneeringly, graphic documentation in the form of thematic maps is always required during a conservation or preservation intervention.

Both for artifacts with greater three-dimensionality and for artifacts with reduced three-dimensionality, the vector graphic drawing is always based on a two-dimensional photographic reproduction of the artifact, which is often not geometrically correct. The artifact is then photographed in all its sides at 360° and a thematic map is created for each side or prospect and the graphic documentation operation is performed during the entire intervention. In the current practice, this process is carried out through manual drawing by restorers, so it is strongly influenced by their skills. 

In these subjective analysis processes, the operations of area graphicization and interpretation of the characteristics constitute a joint phase. Indeed, those who perform the mapping outline the areas of interest directly following the boundaries dictated by their experience and visual perception.

To date, the automatic extraction of drawings from raster images has only been made in archaeology by a specific technique called Stippling. It has been developed to produce illustrations in raster format, extracted from photographs of archaeological objects [3]. Unfortunately, these techniques do not meet the requirements of graphic documentation in restoration. 

Our research aims to study a quantitative analysis methodology to facilitate and semi-automate the recognition and vectorization of areas corresponding to the characteristics under consideration. To this aim, some segmentation algorithms have been tested to separate the characteristics and facilitate identifying the areas and their vectorization. The choice to go beyond the analysis carried out directly on the pixel areas by performing a vectorization was dictated by the fact that the documentation in the restoration of any artifact requires a series of vector drawings, non-illustrative and non-raster, without nuances and with closed polygons topographically consistent with each other. So, the research topic we are developing is not only about one algorithm, but is about a formal methodology involving the cascaded application of a series of algorithms, which has been consolidated over the years. Our work’s novelty is the methodology itself rather than the algorithms used to apply it in practice. This contribution includes the description of the supporting algorithms and the software tools implementing them. The study of it would allow for the full reproducibility of the methodology in practice. 

This methodology has been developed in three years, during which it has been tested on several paintings on canvas, mosaics, frescoes/wall paintings, and paper/parchment artifacts, involving the profiles of diagnosticians, art historians, conservator/restorers and Geographic Information Systems (GIS) professionals. The model presented in this document is the one that has allowed us to obtain the best results in all the tested case studies.

The rest of the paper is organized as follows. In Section 2, the main problems present today in the documentation process are analyzed, and research projects focused on their solution are mentioned. We also mention the graphical documentation software, highlighting those tools that can support our methodology’s semi-automatic implementation, such as, e.g., Geographic Information Systems (GIS). 

In Section 3, we present the various stages of our methodology, which, in Section 4, is applied to a canvas painting. Finally, in Section 5, we conclude the paper and discuss potential future work. 

## 2. State of the Art

Although the importance of documentation has been widely recognized and considerable experience has been gained in applying innovative documentation systems [4], there are still many unsolved problems in analyzing and digitizing artifacts.

### 2.1. Creation of Thematic Maps

Standardized techniques of architectural and archaeological survey (especially those related to the detection of the constituent materials of the external surfaces of buildings) have been the guide for the development of current thematic maps for the planning of conservation, as well as buildings and monuments of all other categories of cultural heritage. For this reason, only in the context of historical architectural monuments and archaeological sites can restorers rely on professionals, e.g., architects, and ad hoc standards for the generation of thematic maps. For interior decorations of buildings and small movable objects such as painted canvases and wood panels, sculptures, and utensils, the conservator-restorer tries to conform to the same architectural survey criteria, often without following a standard methodology. 

The lack of standardization concerns four fundamental aspects: the modalities of photographic acquisition, the modalities of post-production and study of diagnostic investigations, the textual/graphic vocabulary of thematic maps and the use of software to create it. In this section, we focus on using software to create thematic maps and on the study and post-production of diagnostic investigations, because the other two aspects involve complex issues, often dependent on the type of artifact and the regulations in force in each country.

However, we think it is useful to provide some clarification about these issues: the modalities of photographic acquisition change according to the typology of the artifacts and the techniques used to respect the geometric correctness of the artifact represent a very broad field of research. In Italy, the only official document has been drawn up by the Central Institute for Catalogue and Documentation (ICCD) in Rome [5]. Concerning the textual and graphic vocabulary, very few standards are currently in use for digital architectural design [6] and the UNI Beni Culturali NorMal (Norme Materiale Lapideo) standards [1] mainly refer to undecorated stone material, mortars, and ceramics. 

### 2.2. Software in Use

Currently, the most used software and platforms supporting graphic documentation are Computer-aided design (CAD) [7,8] and Geographic Information System (GIS) [9]. An interesting approach is offered in the geographical setting since the problems related to statistical analysis of satellite images and cartography creation are similar to cultural heritage documentation. GIS technologies offer flexible image analysis and data management toolboxes by integrating various functionalities, data types, and formats. In the field of restoration, implementations of GIS were developed in Italy in the 1990s through a project called “Carta del rischio” [10]. GIS and CAD functionalities are also merged into hybrid platforms, like SICAR^®^ [11] a web-based geographic information system officially adopted in 2012 by the Italian Ministry of Cultural Heritage and Activities and Tourism. 

All these tools suffer from some significant practical limitations. CAD is optimal for the mathematical processing of geometric data. However, its use is particularly challenging when the graphic survey to be produced is characterized by irregular and highly jagged edges and shapes. Using CAD drawing tools, operators tend to approximate the area’s perimeters making the edges inaccurate. Furthermore, CAD does not allow for organizing one’s files in a structured database.

Both CAD and SICAR do not allow raster editing; the operator cannot query pixels or optimize the image. Color data are crucial to characterize some types of artifacts, especially those with a decorated or painted surface, of which, for example, the specific conservation problems of each color should be analyzed. 

CAD and SICAR do not allow for any interaction between raster and vector graphics, and each graphic survey is executed by manual drawing. The result is highly subjective, and each thematic mapping is different from any other, even if carried out by the same operator on the same photographic basis. Finally, restorers rarely use a unique system to compile their thematic maps. When dealing with canvases, painted tables, ceramics, fresco paintings, and mosaics, restorers use, often empirically, diverse commercial software for vector graphics and image processing without adopting a standard methodology.

### 2.3. Geographic Information System

In more recent years, the development of low-cost and easy-to-use Geographic Information System software including spatial attributes and mapping elements, has made it possible to use this technology for non-geographic projects. From relatively large areas, GIS has been used on mobile artifacts; especially in Italy, Spain and Portugal, experimentation on the statistical analysis of the degradation of painted canvas and tables has started [12,13,14,15]. Two GIS systems were used for these experiments: QGIS^®^—free and open-source, and ArcGIS^®^, proprietary software of Esri^®^. In particular, QGIS^®^ has proven to be widely used for scientific research in territorial, archaeological, and artistic history. For this software, significant plugins have been developed, among them the Semi-automatic Classification Plugin (Version 7.8.0)that combines multispectral images and raster analysis, allowing the operator-controlled semi-automatic classification of environmental remote sensing images and providing tools to accelerate the process of classification of soil areas [16].

In the archaeological field, a plugin called pyArchInit (Version 2.4.6,) has been developed for QGIS^®^ that allows access to a global database server that can be consulted and modified-like PostreSQL-favoring the homogeneity promoting the homogeneity of the solutions adopted and exporting projects through interactive systems that can be used on the web: the so-called web-GIS [17]. This plugin satisfies the archaeological community’s growing need to computerize excavation documentation, but can also manage documentation in the architectural and historical-artistic fields. 

### 2.4. Diagnostic Investigations

As far as diagnostic investigations are concerned, white balance and color correction, post-production, and interpretation modes are the main issues. 

Digital diagnostic images require a colorimetric correction process, during both acquisition and post-production. In general, all the adjustment and balance operations greatly influence the interpretation of multispectral images as they modify the color tones expressed in the visible spectrum, corresponding to the wavelengths reflected by the material surface. In the case of induced luminescence images that give a response in the field of visible, such as ultraviolet-induced luminescence (UVL) and visible-induced visible luminescence (VIVL) images, artistic materials show different grades and tones of fluorescence according to their composition and aging. Therefore, white balance in shooting and color correction can substantially affect the fidelity and reproducibility of images, making interpretation inaccurate and comparisons between images taken at different times and with different settings inconsistent. As far as IR and UV reflected images and visible-induced infrared luminescence (VIL) images are concerned, white balance mostly impacts on the post-production of false-color reflected images, such as ultraviolet-reflected false color (UVRFC) and infrared-reflected false color (IRRFC).

To obtain consistent and comparable data, it is necessary to follow standards currently represented by the results of the Charisma project of the British Museum [18]. However, these standards are not easy to implement, and in practice, more readily available commercial colorimetric references are used, but they do not offer optimal results. Furthermore, commercially available cameras are designed to provide aesthetically pleasing images and not for scientific analysis of artifacts and, as a result, may make undesired changes to captured multispectral images. The automatic adjustments incorporated into the cameras include white balance adjustments, contrast, brightness, sharpness, Automatic Gain Control (AGC) and control of the dynamic range in low light situations. To solve these problems, in addition to the Charisma Project, several manuals and scientific articles have been published for the correct use of colorimetric references for multispectral imaging [19,20].

The interpretation of multispectral images changes depending on the technique used, the artifact under investigation, and its history dating. Concerning grayscale monochrome images such as infrared-reflected (IRR) and ultraviolet-reflected (UVR), reading and interpretation difficulties depend exclusively on the state of preservation of the artifact under examination and the information searched, so contrast and opacity of the objects in the scene are the only elements on which the restorer and the diagnostician use to recognize the characteristics.

Concerning the phenomenon of reflectance of the materials presented in color diagnostic images, such as induced luminescence images and false-color reflected images, identifying the characteristics searched is more complicated. The characteristics are shown in areas of color that are more or less homogeneous and with variations in tonality and opacity that are always different. These variations depend on the artifact’s execution techniques, the surface materials used and their aging. In addition, the materials, such as pigments and binders, are mixed and layered with each other, and the application techniques vary greatly depending on the historical period. As a result, materials react in multiple spectrum bands simultaneously with different levels of intensity. Therefore, the recognition of characteristics through reading the fluorescence phenomenon is subject to an interpretative process depending on the examiner’s experience and the reference literature for any specific case.

Over the years, many scientific types of research have been conducted in support of the recognition of fluorescent materials. Among the main ones we can mention (a) a mathematical model based on the Kubelka-Munk theory that studies the pigment-binder interaction [21,22], (b) a false color imaging technique called ChromaDI that enhances in the visible image the differences between the optical behavior of the various pigments taking into account the changes that occur during the transition from short to longer wavelengths [23], and (c) a methodology to classify different pigments through Hyper Spectral Imaging (HSI) that acts in the Short Wavelength Infrared (SWIR) region [24]. 

Regarding the statistical analysis of the multispectral image, the same techniques we use here have proved to be very efficient for improving the readability of ancient, degraded manuscripts and palimpsests [25]. Furthermore, in archaeology, these techniques have been used to reveal details otherwise invisible or difficult to discern on the surface of painted walls [26]. However, multispectral techniques are rarely combined with statistical image processing and have never been used to facilitate the restorer in creating thematic maps for documentation. Finally, it is central to specify that regardless of the type of technique and artifact investigated, archaeologists and conservators/restorers are the only ones who know the artifact’s material characteristics and must always be involved in the reading interpretation of diagnostic investigations.

## 3. A New Methodological Approach

In the common documentation practice, the available diagnostic images are analyzed separately and their analysis is based on visual observation of the reflectance phenomena of materials, although in some cases they can be distorted and difficult to interpret. Our strategy instead considers the set of images available as a whole. The aim is to subdivide the painted area into regions through strategies of Blind Source Separation (BSS) [27], which have long been used in other areas, such as document image processing [28]. Then, we propose a rigorous and semi-automated analysis procedure for an easy, fast, and repeatable automatic polygonization of the extracted regions using standard software tools [29]. Therefore, the restorer’s fundamental choice is the only remaining subjectivity, but it is based on a precise and repeatable set of objective and scientifically valid measurements. It is essential to point out that the restorer’s role in directing the investigations and interpreting the images is central and cannot be delegated.

As illustrated in Figure 1, our methodology includes a first phase during which the diagnostic images are acquired and manipulated for analysis purposes. It includes four stages: image acquisition, image segmentation, threshold-based extraction of the regions of interest (ROI), and mapping from raster to vector representation. In the second phase, the methodology applies classification and analysis methods to determine the state of conservation of the artwork. The resulting output is then archived according to the specified requirements. 

### 3.1. Stage 1: Acquisition

The methodology’s efficacy is strongly affected by the quantity and quality of the digital images provided, representing the input for the following stages’ processing and analysis algorithms. As a general rule, the main requirements of photographic reproduction for documentation are precision and consistency with respect to the original dimensions of the artwork, readability of all its parts, high accuracy in color reproduction, uniform illumination on the whole image, and absence of reflections that could impair the analysis. Regarding the respect of the accuracy of the digital reproduction of an artifact, the most used software tools in cultural heritage are Agisoft Fotoscan© and Archis-Siscan©; in particular, Archis is highly useful to the rectification of complex objects or architectures and allows for a two-dimensional graphic restitution of the artifact. Below is a list of the requirements necessary for the methodology we propose.

One of the problems presented by a quantitative approach to the study of artifacts through digital images lies in the strong dependence of obtainable results by changing scale of analysis: indeed, reducing spatial resolution, the information related to the morphology of material features present on the surface is lost. So, for a complete characterization of the survey, it becomes necessary to work on images with a correct spatial resolution, in proportion to the artifact’s size and to the enlargement level on which the analysis will be carried out. Spatial resolution indicates the amount of detail visible in it. As for our methodology, the main advantage consists exactly in the great number of available details, as each detail represents further information. The methodology has been tested on images with different degrees of spatial resolution, we recommend using images with a spatial resolution of 300 pixels per inch—or a minimum of 150 pixels per inch—and 16 bits per channel.

Since the image processing techniques work on multiple images simultaneously, image data coherence is one of the fundamental requirements of the methodology. In order to proceed to the second phase concerning the application of segmentation algorithms, it is necessary that all the images acquired on the same artifact, in the same phase of the conservation/restoration intervention, are correctly registered with each other. Image registration is used to align images of the same subject taken with different acquisition techniques. Alignment involves eliminating slight rotations and tilts and re-sampling the images to the same scale. 

It is not always possible to obtain full alignment of all acquired data because artifacts may change their morphology during the restoration intervention. In these cases, it is essential to provide consistency between sets of data acquired during the same phase of the intervention. Reference [30], dating 2003, aims to present a review of recent and classical image registration methods. More recently, a new image registration framework has been proposed in [31], based on multivariate mixture model (MvMM) and neural network estimation. Reproducibility of the shooting session must be guaranteed by filling out an activity report and a technical diagnostic report, to be kept both to verify the methodology correctness and for further comparisons in time. Reproducibility should also be ensured by saving, cataloguing and archiving all the original files and the connected meta-data, relating to all work phases. 

The choice of the spectral range for image acquisition affects the analysis process, since each range is associated with different quantitative and qualitative information. Concerning the choice of the diagnostic acquisition technique to be applied, this must be assessed by conservators/restorers or by researchers, and will have to be aimed at providing answers to specific (previously expressed) problems, and to questions concerning artifacts conservation and restoration, in their cultural and technological framework. In the absence of multispectral images, the methodology can be applied to images only taken in the visible spectrum. Reference [32] describes an experimental approach to use “color” uniformity alone as a criterion for dividing the image into disjoint regions of interest corresponding to distinguishable features on the surface of the artifact. In detail, this approach was used to highlight overwritten text in a palimpsest, showing that satisfactory results can be obtained with a method of color decorrelation even starting from visible-light images. The results can often be as highly discriminative as those provided by diagnostic and multispectral images. In this case, the color space’s choice is central, as different spaces represent color information in different ways; see, e.g., RGB, HSV, and L*A*B* channels [33,34]. In our tests, the HSV color model has proven to be very useful for color segmentation in complex contexts such as artistic artifacts’ images. The main reason for this usefulness is in V’s components, i.e., in the values corresponding to brightness gradations, which allow to detect even the slightest variations in light intensity, and therefore the smallest discontinuities between areas of interest. Color properties thus described have an immediate perceptive interpretation by conservators/restorers, who are accustomed to distinguish colors according to their visual perception, on which this model is based.

### 3.2. Stage 2: Segmentation

Segmentation is the process of grouping spatial data into multiple homogeneous areas with similar properties. In our case, the properties (or features) we consider are the spectral responses of the different materials, that is, the local reflectance values measured in all the available channels. The regions of interest (ROI) we want to distinguish, segment, and extract are the pixel sets with homogeneous features (such as locations, sizes, and color), which correspond to parts of the artifact made of similar materials. Unfortunately, regions showing different features typically overlap with one another in all the channels, because typically, the materials are mixed and stratified with each other. This often makes segmentation and ROI extraction difficult. The visual inspection performed by diagnosticians or conservators-restorers can be very complicated, time-consuming, or even impossible, particularly when just slightly different spectral characteristics must be distinguished from many channels. Thus, this task can be performed more efficiently and objectively through automated image analysis techniques. In particular, manipulating the input channels to produce a number of maps, each showing a single or a few ROIs, can significantly facilitate the segmentation. Mathematically, this would be accomplished easily if the different materials’ spectral emissions were known, but this is seldom the case. To extract the different regions from multispectral data with no knowledge of their spectra, some assumptions must be made on the regions themselves and the mixing mechanism that produces their spectral appearance. For the mechanism, we assume an instantaneous linear model with M hyperspectral channels and N distinct features
(1)$$x_{i}\left(t\right)=\overset{N}{\underset{j=1}{\sum }}a_{ij}s_{j}\left(t\right),\;i=1,\ldots M$$
where $x_{i}\left(t\right)$ is the value of the data at channel *i* and at pixel *t*, $a_{ij}$ is the spectral emission of the *j*-th feature in the *i*-th channel, and $s_{j}\left(t\right)$ is the value of the *j*-th feature at pixel *t*. Notice that an additional assumption in this model is that the spectra $a_{ij}$ are assumed to be uniform all over the image. This model is also called instantaneous because the data values at each pixel only depend on the feature values in that pixel and not on any neighborhood of it. If we are able to extract the map $s_{j}\left(t\right)$ from the data $x_{i}\left(t\right)$, then it will be easy to extract the ROIs related to the *j*-th feature by just locating the regions where $s_{j}\left(t\right)$ assumes significant values. Extracting $s_{j}\left(t\right)$ from $x_{i}\left(t\right)$ with no knowledge of $a_{ij}$, is a blind source separation problem (BSS), which can only be solved by further assumptions on $s_{j}$. In particular, statistical BSS techniques such as principal component analysis (PCA) [27], and independent component analysis (ICA) [35] reasonably assume that the different features $s_{j}$ have some degree of statistical independence. Indeed, as the patterns formed by different materials in the painted surface are likely to be independent of one another, it is also likely that their central mutual statistics nearly vanish all over the images, that is, assuming zero-mean feature maps:(2)$$\langle s_{k}^{\alpha }\cdot s_{l}^{\beta }\rangle \;≃\;0\;\;\;\;\forall \;k\neq l$$
where *α* and *β* are arbitrary integers, and the angle brackets denote statistical expectation.

Particular cases of (2) are zero-correlation (α = *β* = 1), leading to PCA and other second-order approaches, and statistical independence, i.e., (2) is true for all *α* and *β*, leading to ICA. By these assumptions, the result is obtained by minimizing the following summation with respect to all the $s_{j}$:(3)$$\underset{k\neq l}{\sum }\left|\langle s_{k}^{\alpha }\cdot s_{l}^{\beta }\rangle \right|\;\mathrm{subject}\;\mathrm{to}\;x_{i}\left(t\right)=\overset{N}{\underset{j=1}{\sum }}a_{ij}s_{j}\left(t\right)$$

If the preliminary assumptions are satisfied, this produces a new set of images, each depicting one and only one of the desired feature maps, that is, something approximately proportional to $s_{j}$. By estimating matrix {$a_{ij}$}, both PCA and ICA estimate the features $s_{j}$ by combining linearly the normally correlated $x_{i}$ to produce a different set of images that are uncorrelated or statistically independent. 

Since each output map assumes significant values only where a single feature is present, the ROIs characterized by such a feature can be extracted by just distinguishing between foreground and background. In fact, at best, two primary gray levels dominate each output channel, and only a specific ROI is highlighted. All pixels in the ROI will have similar gray values, and the rest of the image will get confused in the background. 

### 3.3. Stage 3: Binarization

Since the purpose of the methodology is to obtain a precise vector polygon for each classified ROI, the third stage consists of eliminating the overabundance of information caused by the average gray levels, which cause the typical ramp edges between the area of interest and the background. Gradients, in this case, can be classified as noise that slows down and complicates the process of identifying ROIs. To further segment the ROIs from the background, we use a simple recursive thresholding algorithm [36]. 

Given the output channels of the second stage, which correspond to images f(x,y) in grayscale, a gray gradation is fixed, called intensity threshold T. In the binary output image, the pixels labeled with 1 are called object points, while those set to 0 are the background points. The segmentation outcome is strongly influenced by the choice of the threshold T, which can either be constant throughout the image (global thresholding) or vary dynamically from pixel to pixel (local thresholding).
(4)$$g\left(x,y\right)=\{\begin{array}{c} 1,\;\mathrm{if}\;f\left(x,y\right)\geq T\\ 0,\;\;\;\;\;\mathrm{if}\;f\left(x,y\right)<T \end{array}$$

Global thresholding fails in the cases where the image content is not evenly illuminated. Hence, it is essential to respect the lighting consistency during the shooting stage to avoid further optimization pre-processing. The T value can be chosen canonically, using the average value in the grayscale, but in many cases the restorer’s choice is made by trial and error, that is, testing different values to determine one that makes the output satisfactory [37]. This can be done on a single channel and adapted to the others, or by choosing a T value for each channel. In any case, each selected value must be included in the report accompanying the thematic map to guarantee the results’ reproducibility. 

It is worth observing that the choice of the output image to work with is up to the restorer. This choice is needed to filter only the ROIs for the specific thematic map to be generated; in fact, a fully automatic execution would lead to the identification and selection of undesired regions. The final result is a set of binary masks that identify the ROIs by step edges that will facilitate the subsequent stages. In rare cases, the extracted masks may still have noise pixels inside or outside the ROIs. These pixels need to be removed in order not to interfere with the subsequent polygonization processes. For this purpose, basic morphological operations such as region filling, thinning, and thickening can be used to clean noise and modify regions [38].

Alternative approaches to the generation of the binary masks use neural networks, such as, e.g., the Kohonen self-organizing map (SOM) [39,40], which is based on competitive training algorithms [41]. The advantage of SOM is preserving the input samples’ topology [42]. We tested this type of neural network on different types of artifacts, but they proved to be very useful only in some cases, when the ROIs are already visible in all diagnostic images, that is, in cases of easy segmentation. Moreover, the extraction of binary masks with this method proved to be too time-consuming and not easy to use for non-experts. For this reason, we decided not to include them in the final methodology proposed here.

### 3.4. Stage 4: Polygonization

Raster to vector data conversion is a central function in GIS image processing and remote sensing (RS) for data integration between RS and GIS [43]. In general, there are two types of algorithms, namely, vectorization of lines and vectorization of polygons; only the latter is used in our methodology. This function creates vector polygons for all connected regions sharing a communal pixel value [29]. The precision of the mapping depends on several factors, including the spatial complexity of the images. As the resolution of an image becomes sharper, the data volume increases; for this reason, it is important to perform the pre-processing operation of stage 3 to classify information, clean the shapes and the edges of the ROIs, and ensure their topological coherence. 

### 3.5. Methodology in QGIS

In the first phase of the methodology, the raster images are inserted into QGIS using a metric reference system, such as WGS84 EPSG:4326, and associating a worldfile to each image. A worldfile is a collateral file of six plain text lines used by geographic information systems to georeference raster map images. The file specification was introduced by Esri^®^ and consists of six coefficients of a similar transformation that describes the position, scale, and rotation of a raster on a map. This procedure allows us to have the starting data consistent with each other, geometrically correct and divided into different layers according to the acquisition mode.

The second stage involves the analysis of all diagnostic images simultaneously. The PCA algorithm implemented in QGIS is adequate for our purposes—see the Processing Tools for Raster Analysis, in GRASS tools (i.pca), while ICA would require new modules to be implemented in Python. Alternatively, both algorithms can be implemented entirely in Matlab (Version 9.7 R2019b, MathWorks, Natick, MA, USA) [44].

The third stage of binarization is performed individually on each output image. The conservator-restorer chooses among the outputs the ones that best fulfil their cognitive needs. The thresholding algorithm and the morphological functions are present in QGIS, in the list of processing tools of raster analysis, classified for layers or tables. Then, QGIS can perform raster to vector conversion of each binary mask extracted in the third stage. The tool is found in the main menu, in section Raster, Conversion, Polygonization (from Raster to Vector). This conversion occurs very quickly even with complex files and allows for the creation of a vector polygon that faithfully reflects the edges of areas of interest recognized and highlighted by diagnostic investigations.

All the stages of the second phase, related to the operation of classification, analysis, and storage, are performed using computing tools of QGIS to create a database supporting measurements and statistical analysis. Hence, the output is a detailed and personalized description of each polygonal space created, possibly in different file formats, according to the documentation’s needs. Each extracted polygon is classified into categories and subcategories through an attribute table, which favors different analysis types, such as damage assessments and risk evaluation, which can be conducted quantitatively and objectively.

## 4. Case Study

The methodology’s effectiveness is demonstrated in the specific case of a canvas painting by showing that the thematic map that is typically extracted manually can be derived through the stages described above. The chosen artwork is Ecce Homo by Bernardo Strozzi, an oil painting on canvas, 1620–1622, in size 105 × 75 cm^2^, which is in good conservation status. It underwent a restoration in recent times, including a cleaning of the superficial paint layer and the removal of the old restoration interventions; subsequently, a pictorial retouch with paint was carried out. For all phases of our methodology, apart from stage 2 that requires MatLab’s use for the segmentation algorithms, we used the open-source software QGIS (version 3.10.2-A Coruña, whit Grass 7.8.2.).

By following stage 1 of the proposed method, the painting was captured in three different modalities, under visible light illumination (Figure 2a), UV-fluorescence (Figure 2b) and Near-Infrared Reflectography 780–980 nm (Figure 2c). The fluorescence image was subtracted of the visible stray light to highlight the regions that really produce fluorescence under ultraviolet illumination (see [19,45] for details). In the second stage, the three images were processed by PCA and ICA; the output images are shown in Figure 3a,b. A further attempt with ICA has been made on a subset of six channels, obtained excluding the infrared and resulting in the outputs of Figure 3c. It is essential to make explicit that every image produced by the statistical processing no longer corresponds to a specific wavelength range of, but is a recombination of their intensities, highlighting one or more of the ROIs required, which appear in different gray levels. The 20 output images in Figure 3a–c are the new data set that the restorer can inspect for study and feature recognition.

After inspecting the output images, the restorer chose those where the significant regions are most noticeable. These images, reported in Figure 4a,e,i, show some features related to the materials used: Figure 4a highlights the pictorial retouches performed on the background and Figure 4e highlights the pictorial retouches performed on the body of Christ. These two features in the original data were visually superimposed in the same spectral channel, while in this phase they are separated in different outputs, despite having been operated at the same time. Therefore, we can say that segmentation is due to the different pigments used for retouching. In fact, the figure of Christ has likely been restored using titanium white pigment, and the background has been restored using varnish colors. Even without (destructive) chemical analysis or any other more specialized technique, image processing has allowed us to distinguish between regions that look similar in the acquired channels. Analogously, in Figure 4i–l, the red lacquer used for the blood of Christ is highlighted.

The third stage of the methodology, the creation of binary masks through the threshold algorithm, was only carried out on the three outputs chosen by the restorer. In our case, in the range 0–255, we chose a threshold T = 180. Our result is shown in Figure 4b,f,j. For stage 3, we have used the QGIS Plugin Value tool to choose the threshold value, and the raster analysis processing tool Classified for layers or tables for creating the binary masks.

The fourth stage consists of the graphic design’s automatic extraction to create the thematic map. The binary masks shown in Figure 4c,g,k have been transformed from raster to vector drawings, thus obtaining automatically closed vector polygons that comply with topographical rules of adjacency and overlap. For stage 4, we have used the QGIS polygonizer tool from raster to vector.

In the second-phase stages, each extracted polygon is then classified in a corresponding ROI layer and characterized by a different color and texture, see, to create the legend in the thematic map, see Figure 4d,h,l and Figure 5. In particular, for stage 5, we used the characterization in the QGIS Layer Style tool. The extracted polygons corresponding to the ROI have been estimated as a percentage of the total area of interest, dividing them by Layers and associating them into a Table of Attributes. The topological relationship between the database and the graphics is that it is possible to query the data directly by querying the graphic design and automatically exporting the legend and the statistical analysis results in the printing layout phase. QGIS has a useful and versatile Layout Window supporting creating complex sheets and can save templates to be reused in the future. The metadata have been included in the layout, which contains the institution’s logo, the author’s name, the dimensional characteristics of the object, the table number, the date, the documentation operator, and a legend (or glossary) linked to the drawing. This type of metric result is useful for monitoring, conservation, and restoration, see Figure 5. For this case study, we chose to show only the front of the artifact as the back and side profile of the canvas did not have any interesting feature.

## 5. Conclusions

We examined some of the problems concerning the graphic documentation in cultural heritage, i.e., the difficulty in analyzing the diagnostic images, the excessive subjectivity and approximation of the transcription of the relevant information, the complexity, and the long time needed to transcribe information through manual procedures. These problems lead the restorers to choose only the essential information to document, thus making the graphic documentation incomplete and far from guaranteeing reproducibility. There is currently no official or de facto methodological standard that considers all the possibilities offered by image processing and scientific visualization, and commercially available software tools. Consequently, we propose a semi-automated methodology to facilitate and improve the diagnostic investigation while reducing drastically manual interventions. The result of its application is an objective, formal, and accurate graphic documentation to plan restoration, monitoring, and conservation interventions.

Moreover, as a novelty in this field, image segmentation algorithms have demonstrated their potential to reduce subjectivity and accelerate the entire process. The time needed for the entire methodology to be applied can be evaluated according to two factors. The first is the characteristic and speed of the device in use. Basically, all the algorithms used require a short calculation time ranging from a minimum 4–5 s and a maximum of 1–5 min. These times were evaluated considering a spatial resolution of 300 dpi and a low/medium power device. The power of the device’s graphics card and the massive number of images to be examined are the only two factors that can increase the computation time of the BSS algorithms. The second factor includes both the operator’s ability to use the software and the number of thematic maps to be performed. The application of the entire methodology requires a medium/advanced knowledge of the software mentioned. Furthermore, the timing of the creation of the thematic maps depends on the reasoning time of the user themselves and on the features to extract and document.

Based on image analysis processes, the methodology can be applied to any surface, regardless of the spatial complexity of the object or its extent. However, in order to obtain real data, it is necessary that the photographic reproduction of the artefact respects the real dimensions, with the minimum margin of error.

To date, QGIS has been assessed as the most appropriate tool to support each step of the methodology, as in this framework the operator has all the necessary image analysis tools, combining high potential and ease of use. In a previous, empirical and preliminary study preceding the formal methodology [46] we tested the BBS algorithms coupled to neural networks. As future work, we plan to investigate additional algorithms, and in order to ensure a more general applicability of the results we expect to replicate the experiment on a larger number of case studies and the implementation of well-known blind methods to assess the reliability and stability of the results achieved. Another future development could be to integrate our method with some recent experiences and advanced strands of research that are trying to overcome some of the limitations of documentation, offering web-based solutions/platforms able to perform the operations of survey (mapping) of conservation, restoration and preservation in a single environment/system; also by exploiting three-dimensional models [47,48,49,50].

## Figures and Tables

**Figure 1 jimaging-07-00053-f001:**
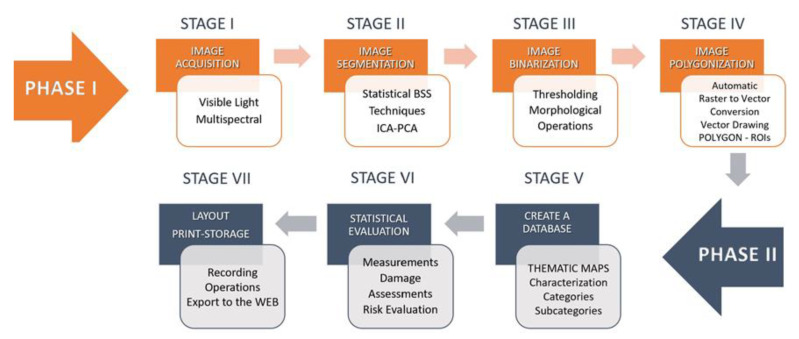
Methodology lifecycle.

**Figure 2 jimaging-07-00053-f002:**
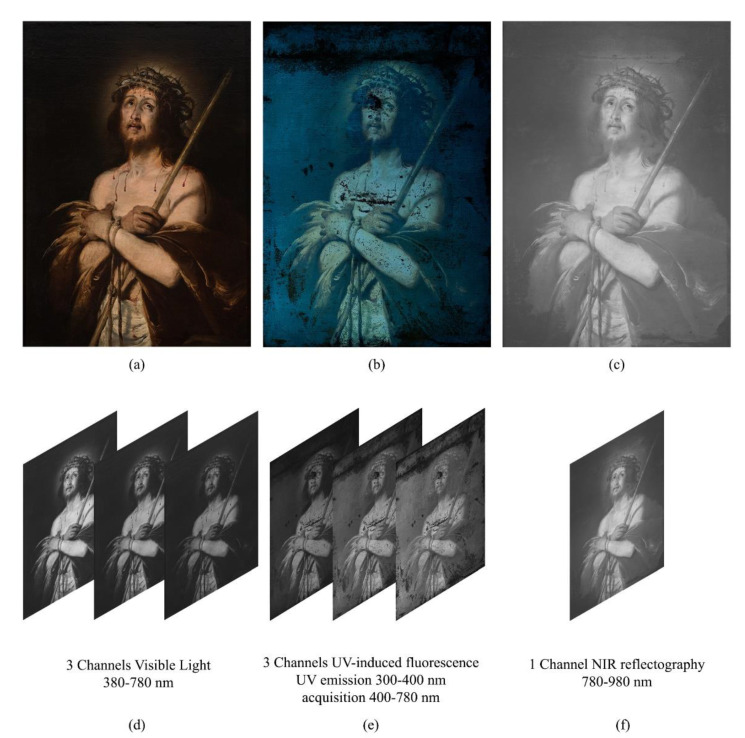
Phase I, stage 1—Ecce Homo by Bernardo Strozzi, oil on canvas, 1620–1622, 105 × 75 cm^2^: (**a**) Standard RGB; (**b**) UV-induced fluorescence; (**c**) NIR reflectography; (**d**–**f**) Respective spectral Channels of the acquired images. Images captured by Paolo Triolo, under permission of the Ministry of Cultural Heritage and Activities and Tourism, National Gallery of Palazzo Spinola, Genova, Italy.

**Figure 3 jimaging-07-00053-f003:**
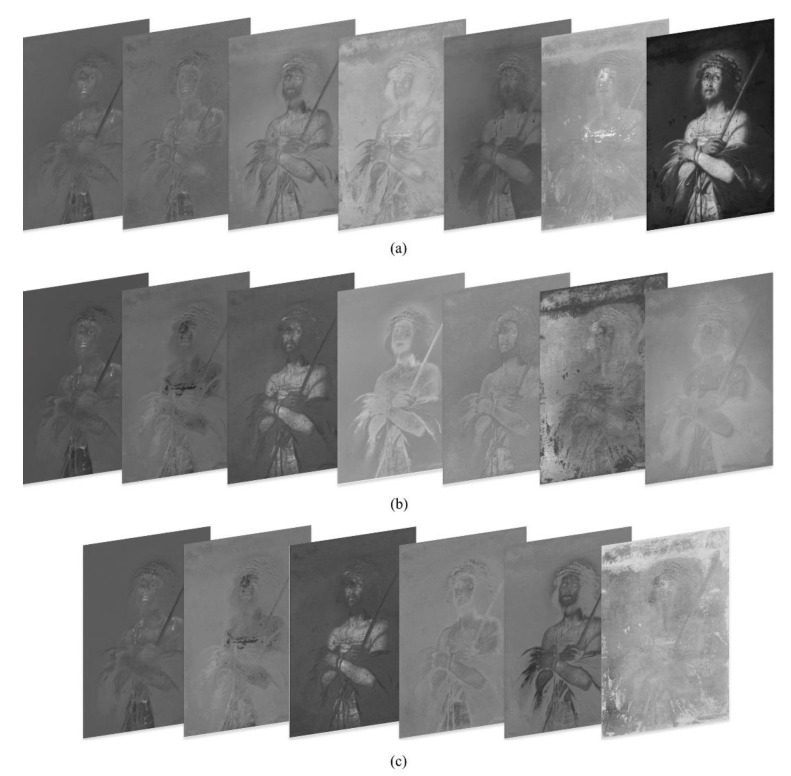
Phase I, stage 2: (**a**) Output channels obtained by principal component analysis (PCA) from the entire data set in Figure 2; (**b**) Output channels obtained by independent component analysis (ICA) from the entire data set in Figure 2; (**c**) Output channels obtained by ICA from the multispectral cube with no IR data in Figure 2a,b,d,e.

**Figure 4 jimaging-07-00053-f004:**
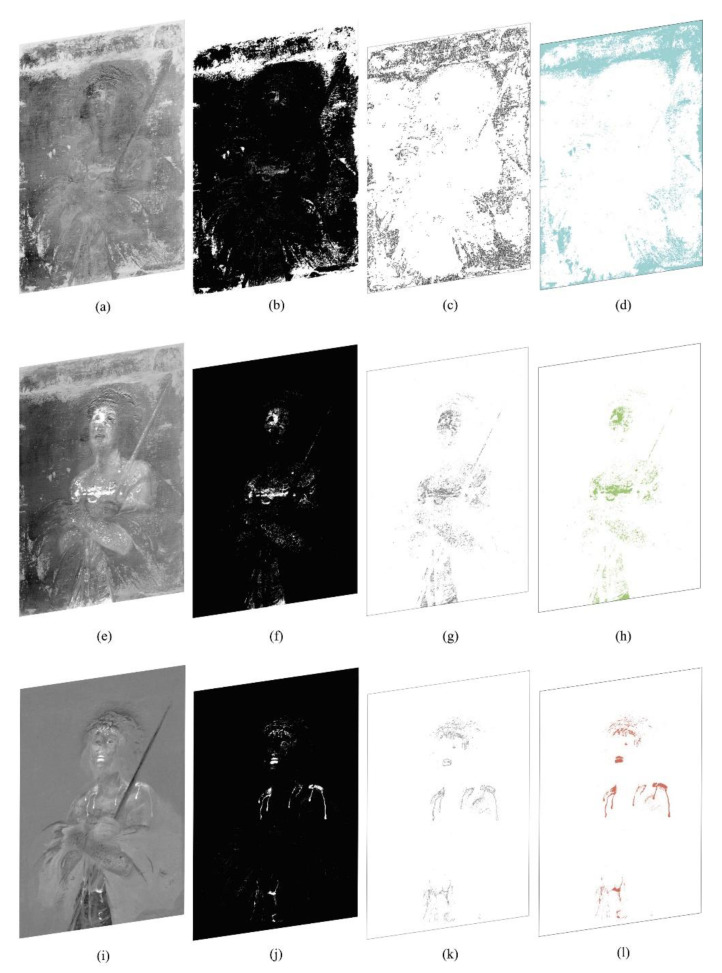
Phase I stages 3–4: (**a**,**e**,**i**) Images processed from the previous stage (in Figure 3) and chosen to identify regions of interest (ROIs); (**b**,**f**,**j**) Corresponding binarized versions; (**c**,**g**,**k**) Image polygonization, raster to vector conversion. Phase II stage 5: (**d**,**h**,**l**) Characterization of the extracted polygons.

**Figure 5 jimaging-07-00053-f005:**
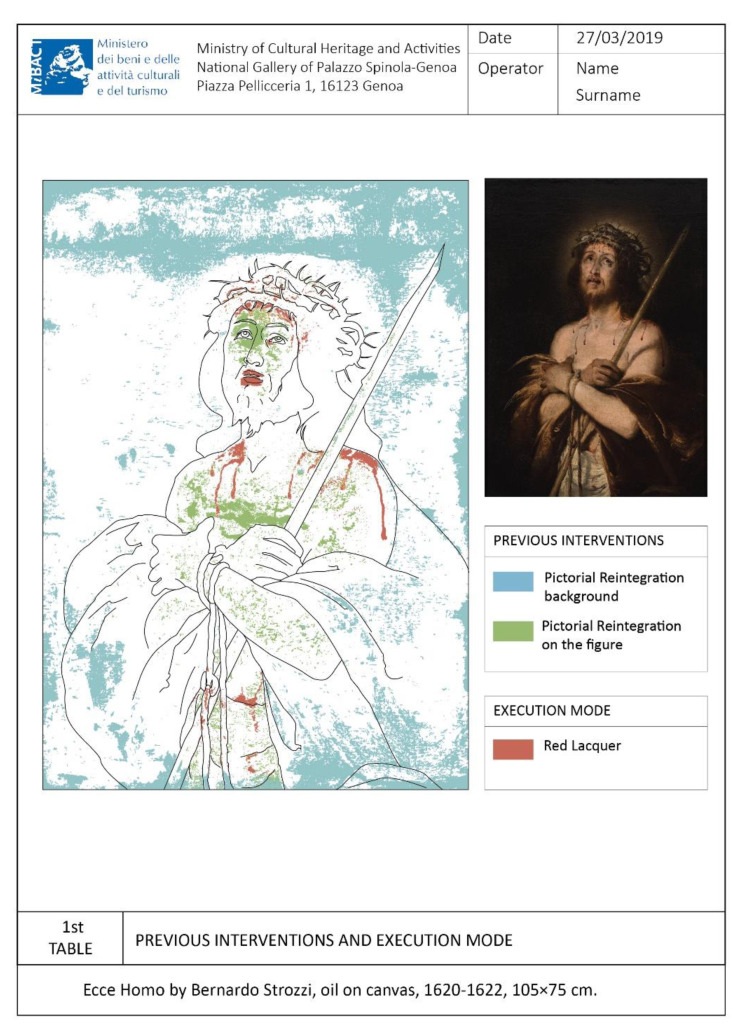
Thematic map-specific colors are assigned to the ROIs (statistical data are omitted).

## Data Availability

The data used to produce the results shown here have been acquired by one of the authors (P.T.) under permission of the Ministry of Cultural Heritage and Activities and Tourism, National Gallery of Palazzo Spinola, Genova, Italy, which maintains all the property rights.

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
