# Peer review of "Analysis of Diagnostic Images of Artworks and Feature Extraction: Design of a Methodology"

_2313-433X, 2021, doi:10.3390/jimaging7030053_

Round 1
Reviewer 1 Report
This paper is well written and provides very useful information. Few comments which I think (if added) may provide more useful information for the reader. 1. In Figure 1 of methodology lifecycle. How long will it take for the entire process? Will it require massive calculation time? 2. In 4. Case Study, example such as The chosen artwork of Ecce Homo by Bernardo Strozzi and the others. How many pixels per image and how many pixels were used in one segment of your imaging processing? What is the adequate range? 3. What is the resolution requirement for the work?Author Response
Thank you for the excellent judgment you gave the manuscript. The questions asked are answered below.
- The time for the entire methodological process can be calculated according to two factors.
- The first factor is the characteristics and speed of the computer in use. Basically, all the algorithms used to require a short calculation time ranging from a maximum of 1-2 minutes to 4-5 seconds. These calculation times were counted starting from a spatial resolution of 300 pixels per inch with a low/medium power calculator. The power of the device's graphics card and the massive number of images to be examined are the only two factors that can increase the computation time of the BSS algorithms.
- The second factor is the operator's ability to use the software and the amount of work he can trust to perform. The application of the entire methodology requires a medium / advanced knowledge of the software mentioned. Furthermore, the timing of the creation of the cards depends on the reasoning time of the user himself and on the features to extract and document.
2-3. In the ECCE HOMO case study we used an image at 300 pixels per inch with a dynamic range of 16 bits per channel. Basically, we would advise against the use of images with low spatial resolution and low dynamic range. The reason is simple: the information in those cases is very scarce and the characteristics are not very visible. We recommend using a minimum of 150 pixels per inch and 16 bits per channel.
Reviewer 2 Report
The paper presents an innovative and interesting methodology aimed at reducing the time and subjectivity during the extraction process (mapping) of the ROI for the purposes of diagnostics of artworks, using multispectral images.
Some aspects, however, should be clarified in order to correctly contextualize the experimentation carried out and its possible/widespread applications.
The proposed methodology, based on the application of different algorithms and SOM neural networks, as documented by the authors, was applied to a single artwork and with the involvement of one restorer involved in the stages of inspection for study and feature recognition.
In order to ensure a more generalized applicability of the results - in consideration of the experimental context conducted - a replication of the experiment on a larger number of case studies and the implementation of well known blinded experiment method would be desirable, that through a significant number of restorers (blinded) it would be possible to verify the reliability and stability of the results achieved.
In the introduction, perhaps, it would be appropriate to clearly indicate how the methodology exposed - on the basis of the elements of caution presented by the same authors in relation to the quality / accuracy of the images acquired, the color space, the spatial registration over time, not applying to artifacts characterized by 3D development - both to be understood as applicable to artifacts with a prevalently two-dimensional development and using images acquired in an indoor environment and according to standard and procedures; these last, to be clearly specified. Herein not clearly presented.
As regards the state of the art, in particular paragraph 2.2, rather than software (referable only to CAD) it would be more appropriate to refer to platforms or information systems, which is to be considered for example SICAR. Although it has to confirm the meaning of the sentence "Finally, restorers rarely use a unique system to compile their thematic maps", however some recent experiences (cited in the present article in the conclusions regarding possible / necessary for future developments) have sought, but above all several advanced strands of research are trying to overcome this limitation by offering web-based solutions / platform capable of allowing the implementation of survey operations (mapping), registration of restoration and maintenance operations in a single environment/system.
Furthermore, a term of unclear understanding is noted: "relief", used in conjunction with "criteria" (r. 92), "Graphic" (r. 111) and also present in the title of the publication cited in Reference no. 43.
The term 'relief' as a noun stands for alleviation, assistance, replacement, art-work; as an adjective it stands for supplementary, etc.
Given the context, it might be that the term "relief" would refer to the term "survey"?
Reference to be checked:
# 42: check 'ana-lisi'
# 18: check availability of Web publication/site: Online: European CHARISMA Project., 2013.
Author Response
Dear Reviewer,
we thank you for pointing out some omissions and lack of clarity in our manuscript. We have added paragraphs in revision mode. Attached you will find a word format file with our answers.

Round 2
Reviewer 2 Report
The additions and changes made improved the article, making it more complete and exhaustive from the point of view of the methodology adopted, solving some small problems / oversights and improving the overall quality.